# The Synergetic Impact of Physical Activity and Fruit and Vegetable Consumption on the Risk of Depression in Taiwanese Adults

**DOI:** 10.3390/ijerph19127300

**Published:** 2022-06-14

**Authors:** Li-Yun Fann, Shi-Hao Huang, Yao-Ching Huang, Chin-Fu Chen, Chien-An Sun, Bing-Long Wang, Wu-Chien Chien, Chieh-Hua Lu

**Affiliations:** 1Department of Nursing, Taipei City Hospital, Taipei 10341, Taiwan; fanly99@gmail.com; 2Department of Nurse-Midwifery and Women Health, National Taipei University of Nursing and Health Sciences, Taipei 11220, Taiwan; 3Department of Medical Research, Tri-Service General Hospital, Taipei 11490, Taiwan; hklu2361@gmail.com (S.-H.H.); ph870059@ndmctsgh.edu.tw (Y.-C.H.); 4School of Public Health, National Defense Medical Center, Taipei 11490, Taiwan; 5Department of Chemical Engineering and Biotechnology, National Taipei University of Technology (Taipei Tech), Taipei 10608, Taiwan; 6Amed Advanced Medication Co., Ltd., New Taipei City 24890, Taiwan; frank.chen@amed.com.tw; 7Center for Technology Transfer and Resources Integration, Fu-Jen Catholic University, New Taipei City 24206, Taiwan; 8Department of Public Health, College of Medicine, Fu-Jen Catholic University, New Taipei City 24206, Taiwan; 040866@mail.fju.edu.tw; 9Big Data Center, College of Medicine, Fu-Jen Catholic University, New Taipei City 24206, Taiwan; 10Taiwanese Injury Prevention and Safety Promotion Association (TIPSPA), Taipei 11490, Taiwan; 11Graduate Institute of Life Sciences, National Defense Medical Center, Taipei 11490, Taiwan; 12Division of Endocrinology and Metabolism, Department of Internal Medicine, Tri-Service General Hospital, School of Medicine, National Defense Medical Center, Taipei 11490, Taiwan

**Keywords:** depression, older adults, leisure-time physical activity, fruit and vegetable consumption

## Abstract

Background: This research focused on the association between physical activity and fruit-vegetable intake and the risk of depression in middle aged and older Taiwanese adults. Methods: Data were obtained from the 1999 to 2015 datasets of the Taiwan Longitudinal Survey on Aging (TLSA), and 4400 participants were included in 1999 (aged ≥53 years). Descriptive statistics provided all of the basic characteristic variables. A chi-square test analyzed the association between sex, age, years of education, marriage, hypertension, drinking, smoking, and the incidence of depression. Logistic regression analysis was used to determine significant associations between physical activity and fruit-vegetable intake, and the presence or absence of depression after 16 years. Results: Combined high physical activity and fruit-vegetable intake reduced the risk of depression by 80% (OR = 0.20, 95% CI = 0.10–0.45, *p* = 0.001) compared to low physical activity and fruit-vegetable intake; high physical activity and moderate or low fruit-vegetable intake caused a 70% reduction (OR = 0.30, 95% CI = 0.15–0.63, *p* = 0.005). High fruit-vegetable intake and low physical activity caused a 65% reduction (OR = 0.35, 95% CI = 0.15–0.63, *p* = 0.005), compared to low physical activity and low fruit-vegetable intake. High physical activity alone caused a 40% reduction, which is the same as by high fruit-vegetable intake alone. Conclusions: Fruit-vegetable intake combined with physical activity was negatively correlated with the risk of depression. More fruit-vegetable intake and physical activity might reduce this risk. The results highlight the importance of physical activity and fruit-vegetable consumption for middle-aged and older adults to prevent depression.

## 1. Introduction

With an aging population and lifestyles changes, Taiwan had an “aging society” in 2018 and will have a “super-aging society” in 2025 [1]. The World Health Organization (WHO) reported that depression is a common mental disorder [2], with an estimated global prevalence of 5% among adults. Depression is characterized by persistent sadness and a lack of interest or joy in previously beneficial or pleasurable activities [2]. The effects of depression can be long-lasting or recurrent, greatly affecting a person’s ability to function and lead a rewarding life [2]. Depression among the older population is generally underestimated [3]. As Taiwan’s population ages, mental illness in the older population deserves and needs attention [4]. The prevalence of depression in older adults is as high as 7–21%, and the rate in females is higher than that in males [4].

Factors causing depression arise from multiple sources, such as biological, physical, psychological, and social [5]. At the biological level, people who are physically active usually have a lower tendency for depression [6]. Fleg proposed that exercise can help reduce the risk of depression [7]. Weng Qikun suggested that participating in exercise can reduce the effect of depression, with low, medium, and high-intensity exercise having a positive effect on depressive symptoms [8].

Several studies have shown that a high intake of fruits and vegetables can also reduce the risk of depression. Increased consumption of fruits and vegetables is associated with a reduced risk of various chronic diseases, including coronary heart disease, obesity, certain cancers, and depression [9]. Some studies show that higher fruit and vegetable intake is associated with a lower likelihood of depressive symptoms [10,11,12].

The SARS-CoV-2 (COVID-19) pandemic has led to rising levels of anxiety, depression and other stress symptoms globally [13]. The occurrence of senile depression will not only cause physical dysfunction and increase the demand for medical care, but even lead to suicide and death, and increase many tangible and intangible costs to family and society [14]. Personality traits, lifestyle, and healthy behaviors in middle-aged and older adults affect their physical and mental health. Physical activity and fruit-vegetable intake are indispensable for healthy behavior patterns [15]. Therefore, we hypothesized that combining physical activity and fruit-vegetable intake can reduce the risk of depression in middle-aged and older Taiwanese adults using the dataset of Taiwan Longitudinal Survey on Aging (TLSA).

## 2. Materials and Methods

### 2.1. Data Sources

This study used survey data of the TLSA (1999 to 2015) provided by the National Health Administration of the Ministry of Health and Welfare [16]. This survey database is representative of the entire Taiwan region, and in the long-term follow-up of a fixed sample group, its survey data has a high completion rate and relocation case tracking. The completion rate of previous surveys is over 80%, which is rare in Asia. 

This database used household registered populations over the age of 60 years who were registered in 311 townships and cities in Taiwan at the end of 1988 as the parent group for sampling, adopting a stratified, multi-stage, random sampling method to select the number of samples. We selected 56 townships and urban areas, sampled neighbors from these selected townships and urban areas, selected two older adults aged over 60 years from each sample of neighbors as interview cases, resulting in a total of 4412 people. In 1989, the first wave of surveys on the physical, mental, social, and health status of middle-aged and older adults in Taiwan was completed. Long-term follow-up was conducted every three to four years. By 2015, six waves of follow-up surveys had been completed. In addition, in 1996 and 2015, in accordance with the sampling method of its baseline survey, a supplementary sample of nationally representative 5066 and 50–56 year old individuals was selected [17].

The survey method used face-to-face interview questionnaires as a data collection tool, collecting information on basic characteristics including family structure, living arrangements, visits of relatives, health status, utilization of medical care, social support and exchange, work, retirement and career planning, leisure and social participation, mood, economic status, and cognition and utilization of social welfare services for the elderly [17].

The fourth wave in 1999 included the assessment of the Elderly Depression Scale (CES-D), which was in line with the content of this study. Therefore, this annual data was used as the baseline of the study to analyze physical activity and fruit-vegetable intake and their effects on depression risk prevention after 16 years [17].

### 2.2. Research Steps and Content

This study included 4400 participants of the TLSA in 1999, >53 years old, excluding those with depressive symptoms (depression scale score: ≥10), diabetes, cancer, BMI < 18.5, death in the following 16 years, and incomplete data (*n* = 752). The included population was 2547 people, and the total effective sample number was 1795. Changes in injury or injury have a greater impact on depressive tendencies than maintainers. A significant correlation was identified between the presence or absence of chronic diseases and the predisposition to depression. In addition, medication or lifestyle changes may be highly associated with depression, and people with low BMI may have other health factors that affect their psychological status. The flowchart from TLSA is shown in Figure 1.

### 2.3. Data Processing and Statistical Analysis

This study analyzed the effect of basic characteristics and combined physical activity and fruit-vegetable intake on the risk of developing depression from 1999 to 2015, according to the following definitions: The dependent variable was depression in 2015, and according to the 2015 questionnaire item (C51), depression (CES-D): “Everyone has mood changes. In the past week, have you experienced any of the following situations or feelings?” The self-reporting of the case is divided into two groups, according to the self-report: 0—no, 1—rarely (only 1 day), 2—sometimes (2–3 days), and 3—often or all the time (>4 days). There were 10 questions in total: (1) I don’t want to eat very much, my appetite is poor; (2) I feel that it is very difficult to do everything; (3) I can’t sleep well (can’t sleep); (4) I feel very bad; (5) I feel very lonely (Lonely, without company); (6) I feel that the people around me don’t want to be nice to me (unfriendly); (7) I feel very sad; (8) I feel unable to do things (no energy to do things); (9) I feel very happy; and (10) I feel that the days (life) are going well. According to the 2007 questionnaire items (C511-11), one of the depression (CES-D) items was (11) I feel that the people around me are not good to me (don’t like me). As this differs from the 1999 report items, it was not included for the sake of comparison. The Depression Degree Index is expressed by the aggregated score of the ten items of the Depression Scale (Center for Epidemiologic Studies Depression Scale, CES-D) according to the Likert Scale method. The total score ranges from 0 to 30, with higher scores representing higher levels of depression. Depression was defined by a 10-point cut-off point [18]. A depression level score < 10 points was classified as "no depression"; while a depression level score ≥ 10 points was classified as "depressed". Independent variables were physical activity and fruit and vegetable intake in 1999. In this study, exercise was defined as leisure time physical activity (LTPA) rather than structured exercise. The exercise situation is multiplied by the exercise frequency, exercise time, sweat volume and whether exercise induced asthma. The calculation method was based on the average weekly exercise frequency (C36), divided into 0×, <2×, 3–5×, and >6×; exercise time (C36b); sweat volume (C36c); and whether asthma was induced on exercising (C36d). The scores were converted into MET (combustion equivalent); exercise groupings were divided into 0–450 METs for low exercise, 450–750 METs for moderate exercise, and >750 METs for the high sports three groups [19]. The frequency of vegetable and fruit intake (CA15a_7 and CA15a_8) was calculated as 0× without food, 0.5×/wk or less, 1.5×/wk, 3–5×/wk, 4× and 6×/d, combined to add up to the weekly intake number. The grouping of vegetable-fruit intake and frequency were divided into three groups of <7, 7–9, ≥10. The exercise situation was combined with the frequency of fruit-vegetable intake, and divided into five categories: low physical activity frequency and low fruit-vegetable intake, high physical activity frequency and high fruit-vegetable intake, low fruit-vegetable intake and high physical activity frequency, high fruit-vegetable intake and low physical activity frequency, and others. Low physical activity and low fruit-vegetable intake were defined as low-intensity physical activity and intake of vegetables and fruits <7×/wk; high physical activity volume and high fruit-vegetable- intake were defined as high-intensity physical activity and fruit-vegetable intake ≥10×/wk (7–9 times); high fruit-vegetable intake and low-physical activity were defined as low or moderate physical activity and fruit-vegetable intake ≥10×/wk; others were defined as only moderate physical activity or fruit-vegetable intake 7–9×/wk or both [19].

Other variables included possible related factors of depression: age, divided into three groups of 53–64, 65–74, and ≥75; years of education were divided into three groups of ≤6, 7–12, and ≥13; marital status was divided into no Occasional and occasional groups; hygiene behaviors including smoking and alcohol consumption were divided into two groups: no and present; BMI was calculated by dividing body weight (kg) by the square of height (meters).

In this study, SPSS 22.0 was used for data processing and statistical analysis. Descriptive statistics were used to analyze the basic variables in 1999. The chi-square test was used to analyze sex, age, educational years, marital status, alcohol consumption, smoking, physical activity frequency, fruit and vegetable intake, and the association of physical activity and fruit-vegetable intake combination with reduced risk of depression. Multiple logistic regression analysis was used to analyze the effects of physical activity and fruit and vegetable intake on the risk of depression over 16 years, respectively, and the two combined conditions. The explanatory variables for this study were as follows: age (53–64, 65–74, and ≥75), educational level (primary ≤6, secondary 7–12, and higher diploma ≥13). The goal of logistic regression was to build an equation that could be used to estimate the probability that an individual would experience a single dichotomous causal (outcome) variable as a function of an independent (predictor) variable. Most real-life phenomena and their corresponding single dichotomous dependent (outcome) variables (e.g., occurrence of postoperative myocardial infarction) are much more complex than can be effectively explained by a single independent variable. Therefore, when there are two or more independent variables identified and included, multiple logistic regression is applied instead. In this study, *p* < 0.05 was considered statistically significant.

## 3. Results

Table 1 shows the sociodemographic variables and univariate variables associated with the risk of developing depression. Marriage, alcohol consumption, physical activity, fruit-vegetable intake, combined physical activity, and fruit-vegetable intake were significant factors (*p* < 0.05). Among the participants, 52.0% were male and 48.0% were female; 55.5% were 53–64 years old, 30.6% were 65–74 years old, and 6.6% were ≥75 years old; 69.7% had ≤6 years of formal education, and the rest had a greater number of years of education. 72.4% had a spouse and 20.3% had no spouse; 72.9% were non-smokers and 19.8% were smokers; 66.9% did not consume alcohol whereas 25.8% did; a high frequency of physical activity was observed in 10.4% participants; the weekly fruit-vegetable intake was <7×, accounting for 9.2%, 7–9×/wk accounting for 9.8%, and >10×/wk accounting for 73.6%; combined physical activity and fruit-vegetable intake: low physical activity frequency and low fruit-vegetable intake accounted for 4.3% of participants, high physical activity frequency and high fruit-vegetable intake accounted for 13.7%, high transport and low fruit-vegetable intake was 2.2%, high transport and low transport accounted for 59.9%, and other combinations totaled 12.5%; the average BMI was 23.96 ± 3.03 kg/m^2^, and the average CES-D total score was 5.26 + 2.42.

Table 2 presents that “high physical activity (34.8%), depression risk (11.6%)” reduced the risk of depression by 40%, which was significant (*p* = 0.001); “moderate physical activity (27.7%), depression risk (15.7%)” decreased the risk by 35%, but was not significant (*p* = 0.257); “fruit-vegetable intake ≥10×/wk (80.0%), depression risk (14.0%)” decreased the risk by 40%, which was significant (*p* = 0.025); “fruit-vegetable- intake 7–9×/wk (10.8%), depression risk (15.3%)” decreased the risk by 25%, but not significantly. Significant difference (*p* = 0.242).

Table 3 presents that under the control of the above variables, “high physical activity and high frequency of fruit-vegetable intake(30.8%), depression risk (10.5%)” significantly reduces the risk of depression by 80% (OR = 0.20, 95% CI = 0.10–0.45, *p* = 0.001); “High physical activity and low Fruit-vegetable intake (3.2%), depression risk (12.5%)” was reduced by 70% (OR = 0.30, 95% CI = 0.15–0.63, *p* = 0.005); “Low physical activity and high intake of fruits-vegetables (40.4%), depression risk (17.5%)” had a 65% lower risk of depression compared with the reference group (OR = 0.35, 95% CI = 0.10–1.05, *p* = 0.038); “Other (15.0%), depression risk (18.0%)” included only or both moderate physical activity and moderate fruit-vegetable intake, with a 67% reduction (OR = 0.33, 95% CI = 0.15–0.78, *p* = 0.015).

## 4. Discussion

The results of our study showed that more physical activity combined with the high intake of vegetables and fruits reduced the risk of depression in a duration of 16 years by 80%. “High fruit-vegetable intake combined with low physical activity” also reduced the risk of depression by 65%, also rarely suffer from the risk of depression by 70%. “Other” reduced the risk of depression by 67%. In the case of no combination, only high physical activity or only consuming fruits and vegetables ≥10×/wk reduced the risk; however, moderate physical activity or 7–9×/wk fruit-vegetable intake showed no significant results. The main variables that affect the occurrence of depressive tendencies.

### 4.1. The Association between Physical Activity and Depression Risk

Physical activity and intake of fruits and vegetables were the main variables affecting the risk of depression. Aihara et al. found that older adults in the Japanese community who had physical activity habits, a daily balanced diet, a steady intake of dairy products, and fixed interests were less prone to depression [20]. Xu et al. identified that regular exercise substantially helped mental health, helping to reduce depression, stress, and anxiety, and improved mood, enhanced happiness, and improved quality of life [21]. The greater the frequency and time of physical activity, the lower the effect of depression [22]. In addition, while exercising, attention is diverted and troubles are temporarily forgotten; after exercising, the whole body feels comfortable and the effect of relaxation is achieved, which is helpful for the treatment of depression [23]. In China, Peng Yuren and Zhang Shuling identified that both aerobic or anaerobic activity had an obvious effect on reducing anxiety and depression [24], Wu Jiabi and Liu Zhaoda agreed that regular physical activity could distract or divert attention, thereby reducing anxiety and improving unpleasant thoughts during the activity, which is often regarded as non-drug adjuvant therapy or alternative therapy (adjuvant or alternative therapy), and has no side effects [25]. Studies by Shi Chunhua and others showed that people with physiological diseases, such as chronic diseases, malignant tumors, or other illnesses tended to be more prone to depression [26]. Sherri suggested that long-term regular exercise and proper physical activity can improve the overall health and physical fitness of the person and also reduce chronic diseases such as cardiovascular disease, hypertension, diabetes, breast cancer, and osteoporosis and prevent the occurrence of heart disease and stroke [27]. In terms of the psychological state, it could reduce depression and anxiety and improve mood and emotion regulation [28]. These studies have shown that the amount of exercise habits can reduce the risk of depression and also prevent other chronic diseases. Our study found that individuals with “high physical activity combined with low fruit and vegetable intake” and “high physical activity” are only associated with a lower risk of depression than those who do not physical activity on weekdays 70% (*p* = 0.005) and 40% (*p* = 0.001).

### 4.2. Association of Fruit and Vegetable Intake with Depression Risk

You Ruifeng and Cai Zhonghong conducted a study on the Relationship between Food Intake Frequency and Depression Risk in Taiwanese Elderly People and found that Taiwanese older adults who consumed fruits-vegetables ≥3x/wk had a reduced risk of depression of 38%, compared with those who consumed less fruits and vegetables [29]. Another study by Akbaraly et al. found that those who ingested non-processed foods high in fruits, vegetables, and fish (whole food) had a lower risk of depression than those who ingested excessively processed foods such as fried food, desserts, and refined grains [30]. Nanri et al. conducted a study with Japanese people and found that when the intake was divided into three levels, those with the highest intake of vegetables, fruits, mushrooms, and soy products had a 56% lower risk of depression than those at the lowest level [31]. Sanchez-Villegas et al. conducted a follow-up survey of middle-aged Spanish adults and showed that a Mediterranean diet based on vegetables, fruits and grains, and beans and fish helped reduce the incidence of depression [32]. Woo et al. investigated the association between nutrient intake, cognition, and depression in older Chinese adults and found that vitamin A, B, and C, fiber, and vegetable intake were negatively correlated with depression [33]. Oishi et al. explored the relationship between nutrition and depressive symptoms in Japanese adults aged 65–75 years [34]. Karakula et al. [35] reviewed the literature on diet and mood from 1990 to 2007. The results showed that insufficient intake of vitamins B6 and B12, folic acid, and omega-3 fatty acids increased the risk of depression [35]. Benton and Donohoe proposed that the intake of vegetables negatively correlated with depression, and plant-based foods increased the release of endorphins in the brain to improve mood. These studies showed that intake of vegetables and fruits reduced the risk of depression [36]. The results of our study showed that the risk of depression was 40% lower in those with a “high intake of vegetables and fruits” than those with a “low intake of vegetables and fruits” (*p* = 0.025). The risk of developing depression was reduced by 63% (*p* = 0.048), which was consistent. Furthermore, “7–9×/wk of fruit-vegetables intake” also reduced the risk of depression by 25%, but not significantly (*p* = 0.242).

### 4.3. Association between Nutrition, Physical Activity, and Depression Risk

The exact mechanism by which fruits-vegetables lead to a reduced risk of depression has not been precisely identified. However, there is some evidence for associations between the nutrients (e.g., magnesium, zinc) and antioxidants (e.g., vitamins C, E, and folic acid) found in these foods [37]. One possible pathway involves folate, a common vitamin found in foods such as leafy green vegetables, beans, legumes, and citrus fruits. Folic acid plays a key role in the regeneration of tetrahydrobiopterin (BH4) and remethylation of homocysteine, leading to the production of S-adenosylmethionine (SAMe) [37]. In addition, both SAMe and BH4 are important cofactors in the production of neurotransmitters such as serotonin, dopamine, and adrenaline, all of which play crucial roles in mood regulation. Folate deficiency has been associated with depression [37]. Physical exercise is reportedly an effective way to relieve depression [38]. Regular physical activity increases beta endorphin production [39]. Under the stimulation of endorphins, adolescents are in a relaxed and happy physical and mental state [40]. Endorphins are also known as “happiness hormones” or “youth hormones”, which means that they influence happiness in teenagers and youthfulness in older people [41]. Physical exercise of appropriate intensity can accelerate blood circulation to various organs of the body, promote metabolism, and relax muscles to ease the mind [42]. Physical exercise can regulate the secretion of neurotransmitters in the human body. Typically, neurotransmitters, such as adrenal cortex, serotonin, and dopamine, promote positive emotional experiences of excitement and pleasure in individuals, improving resistance to depression [43]. A certain concentration of neurotransmitters is required for them to be effective. High-frequency, long-term physical activity ensures the accumulation of neurotransmitters in the brain and lays the foundation for mood enhancement and depression relief in adolescents [44]. Evidence from a meta-analysis study indicated a positive association between physical activity levels and low depression levels [45]. The results of our study showed that more physical activity combined with high intake of vegetables and fruits could reduce the risk of depression over a duration of 16 years by 80%. “Low physical activity combined with high intake of vegetables and fruits” also reduced the risk of depression by 65% and by 70% The "Other" group rarely suffer from the risk of depression by 67%. In the case of no combination, only high physical activity or only consuming fruits and vegetables ≥10×/wk can also effectively rarely suffer from the risk, but moderate physical activity or 7–9×/wk fruit-vegetable intake was not significant. The main variables that affect the occurrence of depressive tendencies.

### 4.4. Other Findings

In the analysis of the relationship between health behavior and depression, depression in women aged 45–64 years was slightly related to alcohol consumption [46]. Japanese studies found that women who consumed alcohol moderately and maintained good relations with neighbors were less likely to have depression. This is consistent with the results of this study, as those who consume alcohol had a significantly lower risk of depression than those who did not (*p* = 0.001). Lv Shuyu and Lin Zongyi investigated the older-aged community in Kaohsiung City [47]. The results showed that age was one of the factors related to depression symptoms, and the older the age, the greater the tendency to depression [47]. Wang Peiqi [48] pointed out that people with no spouse were more prone to depression. The reason may be the increase in factors such as knowledge and experience with age. With improved policies or a sound retirement system, the elderly may be more relieved regarding factors that may cause depression. Other protective factors include higher education and socioeconomic status, participation in worthwhile activities, and religious or spiritual participation [49]. Treatments including behavioral therapy, cognitive behavioral therapy, cognitive literature therapy, problem-solving therapy, brief psychodynamic therapy, and life review/recall therapy are effective [49].

In summary, the above studies have shown that more frequent intake of fruits and vegetables and regular physical activity reduced the risk of depression. This study observed that only vegetable and fruit intake 7–9×/wk or moderate physical activity can help reduce the risk of depression. However, no group reached statistically significant results.

### 4.5. Research Limitations

Limitations of this study include the following: (a) The inferences and references may not be suitable for young people because the research subjects were middle-aged and older adults; (b) The number of chronic diseases and comorbidities in the elderly may affect physical and mental health, resulting in the occurrence of depression, which was not discussed in this research. Further research including these aspects is recommended; (c) The database used included interview data that was mostly self-reported, and self-reported data inevitably has problems with accuracy, such as errors caused by aging-related cognitive degradation or memory loss; (d) The food intake referenced in this study was obtained by inquiry and estimated according to the food frequency questionnaire (FFQ) but cannot be fully quantified and (e) Lastly, the longitudinal analysis covers a relatively long period between two surveys that merely reflect two snapshot conditions. It is possible that depressive symptoms, fruits and vegetables consumption, and physical activity could have changed over time.

## 5. Conclusions

We conducted a study on the relationship between physical activity combined with fruit-vegetable intake on the risk of depression in Taiwanese middle-aged and older people. Our results showed that the combination of physical activity and intake of more fruits and vegetables can effectively reduce the depression risk in this population. Physical activity and fruit-vegetable intake are important healthy behaviors which are safe, effective, cost-effective, and can help middle-aged and older adults maintain good physical and mental health and reduce the risk of depression. This result highlights the importance of exercising more and consuming more fruits and vegetables during weekdays for middle-aged and elderly people. This healthy behavior is safe, effective, and cost-effective, and can reduce national medical expenses. However, in addition to exercising more and consuming more fruits and vegetables on weekdays for middle-aged and elderly people, the government or non-governmental welfare organizations must also provide appropriate social welfare services, medical insurance, and education and social activities for the elderly to promote the successful aging of the elderly so that they can enjoy a happy and joyful old age.

## Figures and Tables

**Figure 1 ijerph-19-07300-f001:**
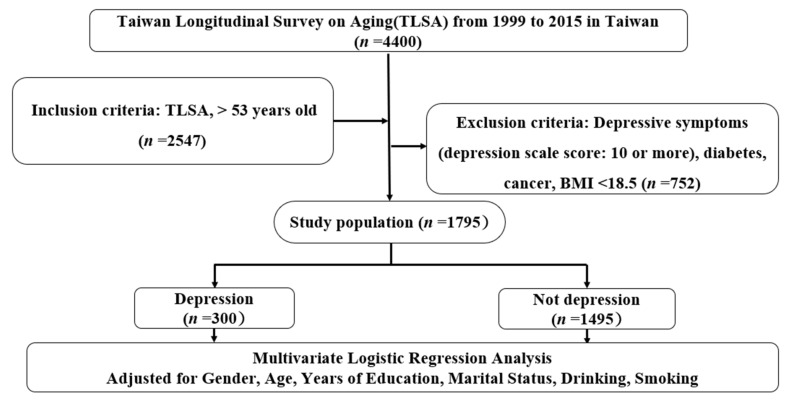
Flowchart of this study.

**Table 1 ijerph-19-07300-t001:** Correlation between lifestyle and health-related variables of the research sample from 1999 to 2015 was analyzed using the chi-square test *(n* = 1795).

Item	Overall, *n* (%)	Depression, *n* (%)	Not Depression, *n* (%)	*p*-Value
**Sex**				**0.789**
Male	933 (52.0)	165/933 (17.6)	768/933 (82.4)	
Female	862 (48.0)	135/862 (15.6)	727/862 (84.4)	
**Age (mean ± SD, year)**				**0.876**
53–64	996 (55.5)	145/996 (14.5)	851/996 (85.5)	
65–74	549 (30.6)	107/549 (19.5)	442/549 (80.5)	
≥75	118 (6.6)	50/118 (42.4)	68/118 (57.6)	
**Years of education**				**0.687**
≤6	1252 (69.7)	205/1252 (16.3)	1047/1252 (83.7)	
7–12	299 (16.7)	65/299 (21.7)	234/299 (78.3)	
≥13	112 (6.2)	39/112 (34.8)	73/112 (65.2)	
**Marital status**				*** 0.039**
Married	1299 (72.4)	215/1299 (16.5)	1084/1299(83.5)	
Unmarried	364 (20.3)	85/364 (23.3)	279/364(76.6)	
**Smoking status**				**0.078**
Yes	355 (19.8)	63/355 (17.7)	292/355 (82.3)	
No	1308 (72.9)	229/1308 (17.5)	1079/1308 (82.5)	
**Alcohol consumption**				**** 0.001**
Yes	463 (25.8)	67/463 (14.5)	396/463 (85.5)	
No	1200 (66.9)	235/1200 (19.5)	965/1200 (80.5)	
**Amount of physical activity**				**** <0.001**
Low	1029 (573)	135/1029 (13.2)	894/1029 (86.8)	
Moderate	445 (24.8)	80/445 (17.9)	355/445 (82.1)	
High	186 (10.4)	85/186 (45.6)	101/186 (54.4)	
**Fruit and vegetable** **intake frequency (times/week) ^a^**				*** 0.025**
<7	166 (9.2)	35/166 (21.1)	146/185 (78.9)	
7–9	176 (9.8)	32/176 (18.1)	144/176 (81.9)	
≥10	1321 (73.6)	214/1321 (16.1)	1107/1321 (83.8)	
**Physical activity and fruit and vegetable intake**				*** 0.043**
Both low	77 (4.3)	11/77 (14.2)	66/77 (85.7)	
Both high	246 (13.7)	72/246 (29.2)	174/246 (70.7)	
High Transport, Low fruit-vegetable intake	40 (2.2)	8/40 (20.0)	32/40 (80.0)	
Low Shipping, High fruit-vegetable intake	1075 (59.9)	152/1075 (14.2)	923/1075 (85.8)	
Other	225 (12.5)	56/225 (24.8)	169/225 (75.2)	
**B** **MI (kg/** **m** ** ^2^ ** **)**	23.96 + 3.03	23.95 + 3.02	23.97 + 3.03	**0.882**
**CES-D overall Score ^b^**	5.26 + 2.42	13.30 + 3.49	5.37 + 2.57	**** <0.001**

^a^ Low in both items means low-intensity physical activity and fruit-vegetable intake <7×/wk; high in both items means high-intensity physical activity and fruit-vegetable intake ≥10×/wk (7–9×); low physical activity and high fruit-vegetable intake ≥10×/wk with low or moderate physical activity; other indicates only or both moderate physical activity or 7–9×/wk of fruit-vegetable intake. ^b^ Mean + standard deviation. *: *p* < 0.05, **: *p* < 0.01.

**Table 2 ijerph-19-07300-t002:** The effects of physical activity and fruit-vegetable intake at the benchmark point on the risk of depression after 16 years, analyzed by logistic regression analysis (*n* = 1795).

Depression Risk from 1999 to 2015				
Variable ^a^	Total (%)	Depression Risk (%)	OR (95%CI)	*p*-Value
**Amount of physical activity**				
complicate	37.5	19.2	1	
Moderate	27.7	15.7	0.65 (0.23–0.78)	0.257
high	34.8	11.6	0.60 (0.42–0.78)	** 0.001
**Fruit and vegetable intake (times/week)**				
<7	9.2	20.0	1	
7–9	10.8	15.3	0.75 (0.54–1.06)	0.242
≥10	80.0	14.0	0.60 (0.40–0.78)	* 0.025

OR = odds ratio; 95% CI = 95% confidence interval. ^a^ This model depends on whether there was a risk of depression from 1999 to 2015; the model controls variables such as sex, age, years of education, marital status, smoking status, and alcohol consumption since 1999. *: *p* < 0.05, **: *p* < 0.01.

**Table 3 ijerph-19-07300-t003:** Logistic regression analysis of the effects of combined physical activity and fruit-vegetable intake on the risk of depression after 16 years (*n* = 1795).

Depression Risk from 1999 to 2015				
Variable ^a^	Total (%)	Depression Risk (%)	OR (95%CI)	*p*-Value
**Physical Activity and Fruit and Vegetable Intake ^b^**				
Both are low	1.5	35.7	1	
Both high	30.8	10.5	0.20 (0.10–0.45)	** 0.001
High physical activity, Low fruit-vegetable Intake	3.2	12.5	0.30 (0.15–0.63)	** 0.005
Low physical activity, High fruit-vegetable Intake	40.4	17.5	0.35 (0.10–1.05)	* 0.038
Other	15.0	18.0	0.33 (0.15–0.78)	* 0.015

OR = odds ratio; 95% CI = 95% confidence interval. ^a^ This model depends on whether there is a risk of depression from 1999 to 2015; this model controls variables such as sex, age, years of education, marital status, smoking status, and alcohol consumption since 1999. ^b^ Low in both items means low-intensity physical activity and intake of vegetables and fruits <7×/wk; high in both items means high-intensity physical activity and intake of vegetables and fruits ≥10×; High sports and low vegetables are high exercise and vegetable and fruit intake <7 or 7–9 times per week; low physical activity and high fruit-vegetable intake ≥10×/wk and low or moderate physical activity; others indicates only or both moderate physical activity or 7–9×/wk of fruit-vegetable intake. *: *p* < 0.05, **: *p* < 0.01.

## Data Availability

Data are available from the Ministry of Health and Welfare (MOHW) provides a series of databases for the research and analysis of the “Long-term Follow-up Survey on Physical, Mental and Social Conditions of the Middle-aged and the Elderly”. Because of legal restrictions imposed by the government of Taiwan concerning the “Personal Information Protection Act”, data cannot be made publicly available. Requests for data can be sent as a formal proposal to the MOHW.

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
