# Peer review of "The Synergetic Impact of Physical Activity and Fruit and Vegetable Consumption on the Risk of Depression in Taiwanese Adults"

_ijerph, 2022, doi:10.3390/ijerph19127300_

Round 1

Reviewer 1 Report

Above, the current manuscript is very poor in writing. The whole manuscript meeds to be revised by a native speaker. Some of the sentences are extremely hard to read. For example, the very first sentence takes 5 lines and is not correct in grammar.  There are also obvious typos and wrong sentences throughout the manuscript. For example, in line 68, there are two % signs, and the following sentence is also not correct in grammar.  In line 383, the authors used odds ration instead of odds ratio. 

The unpolished manuscript should have not been submitted and the language barrier has prevented me from diving into the research itself. 

Table 1 and Table 2 should be combined, with columns of depression group, non-depression group and overall. The authors made no distinction between risk/prevalence and odds when interpreting results from logistic regressions.  It is hard to believe that physical activity and diet pattern in 1999 can predict depression up to 2007. 

Reviewer 2 Report

Thank you for the opportunity to peer review the valuable manuscripts.

I agree with your opinion that physical activity and nutrition for depression in middle-aged and elderly individuals are important. However, please consider the following points to clarify this study.

What is new and unique about this study?  Previous studies have shown that nutrition and physical activity significantly affect depression. However, discussion of this point is limited and merely a review of previous studies. Please correct it in the Introduction and Discussion sections.

In the Methods section, please specify the selection criteria for the logistic regression analysis factors. How were age and educational years included in the logistic regression analysis? If categorical variables were used in the results, please modify and include real numbers.

In the Results section, the texts in the tables are misaligned and have unnecessary spaces, which are difficult to read; therefore, please organize them. Additionally, the explanations in the tables and texts are confusing and difficult to read; therefore, please correct this.

In the Discussion section, the statement “The results of this study showed that more exercise combined with high intake of vegetables and fruits can effectively reduce the risk of depression in the following eight years by 75%,” is a different expression. Presently, it is a prospective observational study, not an interventional study. The observation revealed that there were fewer patients with depression in the group with high frequency of exercise and vegetables; therefore, it is more accurate to state that “there were few depressed subjects” than “there were reduced depression subjects.”

The reasons stated in the Discussion section in page 11 lines 505–508 do not explain the difference of conclusions from previous studies. It is only a speculative opinion; therefore, the author should provide evidence to support the data of the study and other studies and opinions.

Round 2

Reviewer 1 Report

The authors have improved the manuscript in writing significantly since last submission. The overall structure is great. Unfortunately, most of the scientific concerns are not addressed clearly:

1. The distinction between odds and risk is still somewhat unclear.In the updated table one, for example, the two column should have been depression vs. not depression rather than risk of depression vs. not risk of depression. Same issue for figure 1. Also in line 324, how is it possible that risk reduced by 80% when odds ratio is 0.2? The authors need to clearly understand their reported effect sizes, otherwise the conclusions thereafter are not convincing.

2. Another major issue lies in the study design: The model tried to prediction depression in 2015 (line 151) based on exercise and fruit intake in 1999 (line 171, 172).  Could individuals' exercise and diet pattern have been changed in the 15-year period that eventually leads to their outcome in 2015?  The current analysis is based on the strong, albeit unlikely true, assumption that exercise and diet pattern remains unchanged since 1999 up to the time individual was diagnosed of depression before 2015. Also, is it possible that there were individuals who was depressed but recovered in 2015? 

Reviewer 2 Report

Thank you for your response. 

If the report as described to me is correctly reflected in the manuscript, this is fine.
